# Utilizing Piezo Acoustic Sensors for the Identification of Surface Roughness and Textures

**DOI:** 10.3390/s22124381

**Published:** 2022-06-09

**Authors:** Kayra Kurşun, Fatih Güven, Hakan Ersoy

**Affiliations:** 1Department of Mechanical Engineering, Akdeniz University, Antalya 07070, Turkey; kayrakursun@akdeniz.edu.tr; 2Department of Machinery and Metal Technologies, Başkent OSB Vocational School of Technical Sciences Hacettepe University, Ankara 06909, Turkey; fatihguven@hacettepe.edu.tr

**Keywords:** roughness, machining, acoustic, measurement

## Abstract

This study examines surface roughness measurements via piezo acoustic disks and appropriate signal processing. Surface roughness is one characteristic of surface texture that can have various irregularities inherent to manufacturing methods. The surface roughness parameters and corresponding surface profiles are acquired by a stylus profilometer. Simultaneously, elastic waves propagated along metal surfaces caused by the friction of a diamond tip are obtained in the form of raw sound via piezo acoustic disks. Frequency spectrum analysis showed apparent correlations between the traditionally obtained measurement parameters and the piezo acoustic measurement data. Thus, it is concluded that acoustic friction measurement shows promising results as a novel measurement method for the surface roughness states of certain materials.

## 1. Introduction

The surfaces of machine parts often contain irregularities that result from manufacturing methods. Surface roughness, one such irregularity, is one of the important parameters used to describe surface texture. Surface roughness can cause nonlinear problems in fluid systems, especially in machine parts that contact each other. Studies have shown that surface roughness affects contact pressure in interference fit [1,2], preload loss in bolted connections [3,4], surface wear properties [5,6], surface coating properties [7], the fatigue strength of machine parts against dynamic stresses [8,9], the strength of adhesive-bonded joints [10,11], and frictional losses in fluid systems [12].

Surface roughness measurement is defined in ISO 21920-2:2021 [13] and has different parameters. *Ra* is the most commonly used parameter in determining the surface texture state, as it provides the most general and adequate information about the surface roughness. However, when the fluctuations are sudden and large in amplitude, the sensitivity decreases. Therefore, it is a better option to use the *Rz* parameter when sensitivity becomes important. Because the mathematical formulations depend on the five highest peaks and the five smallest valleys in the surface profile, they capture sudden spikes more accurately [14]. In this study, the main goal is to establish a correlation between acoustic measurements and surface roughness measurements. Therefore, the comparison with *Ra* in terms of the general condition of the surface roughness and the comparison with *Rz* in terms of sensitivity have gained importance in terms of determining which parameter is more proportional to the acoustic measurement values. Therefore, both parameters are considered within the scope of this study.

Surface roughness measurements are generally divided into contact and noncontact methods. Stylus profilometry, a contact measurement method, is generally performed by moving a diamond tip over the surface. Noncontact methods generally comprise measurement methods that optically monitor the surface. However, in these methods, the surface to be measured must be exposed and accessible. Laser profilometry is a highly preferred noncontact method. In this method, the laser beam intensity is dependent on the material surface brightness. In this way, the surface roughness profile can be mapped. The roughness mapping process plays an important role in health and pharmaceutical sciences, where two-dimensional data acquisition is critical [15]. In addition, modulalography is a special-case surface mapping method that is used for reconstructing Young’s modulus of coronary arteries from radial-strain ultrasound images [16].

Surface roughness studies remain one of the most interesting topics in mechanical engineering. In addition to traditional methods, new surface roughness measurement methods continue to be developed [17]. These studies also note how important it is to measure surface roughness values accurately, quickly, and effectively. Although acoustic wave sensors and technologies have been used in certain engineering system measurements, their use to define surface roughness has only recently attracted attention [18,19].

Through the acquisition of acoustic or elastic wave propagation within solids, various physical data can be obtained and analyzed accordingly. Related studies can be seen in the literature as follows: the analysis of cutting parameters of machine tools with acoustic measurements [20,21]; tool life prediction via sound frequency analysis [22,23,24], fault detection in roller bearings by wavelet packet transform [25], in train bearings by improved harmonic product spectrum [26], and in low-speed bearings using acoustic emission sensors [27]; plate fatigue evaluation using high-mode Lamb waves [28]; investigation of the effects of elastic waves on fractures [29]; and surface stress measurement with Rayleigh wave detection [30]. In addition, while low-cost lead zirconate titanate (PZT) transducers were previously used as buzzers for random sound generation, there are also studies where they are used as acoustic sensors [31] for structural health monitoring [32]. Moreover, PZT transducers and acoustic emission (AE) sensors are also used together to measure the surface quality in the grinding process, and it was determined that the AE sensor and PZT transducer gave similar responses to the stimuli from the grinding process [33]. In a similar study in which AE sensors and PZT transducers were compared by applying the PLB-(pencil lead break) test on alumina ceramic material, it was determined that the obtained frequency values showed that PZT could be used as an alternative to AE sensors [34].

In this study, the measurement and calculation of the *Ra* and *Rz* surface roughness parameters are first investigated. Subsequently, the surface wave propagation equations are defined. The *Ra* and *Rz* values of the prepared test specimens are then measured. Finally, all obtained data are transferred to a computer environment to be compared and analyzed.

## 2. Materials and Methods

### 2.1. Surface Roughness Equations

Surface roughness parameters are calculated according to ISO 21920-2:2021 [13]. The average absolute value (*Ra*) is shown in Figure 1 and calculated according to Equation (1).
(1)$$Ra=\frac{1}{l_{n}}\int_{0}^{l_{n}}\left|z\left(x\right)\right|dx$$
where *l_n_* and *z*(*x*) are the relative length and amplitude function of the profile, respectively.

The average profile height (*Rz*) is the average of 5 heights, as shown in Figure 2, and calculated according to Equation (2).
(2)$$Rz=\frac{1}{5}\left(Rz_{1}+Rz_{2}+Rz_{3}+Rz_{4}+Rz_{5}\right)$$

### 2.2. Governing Equations for Sound Propagation on Solid Surfaces

Surface Sound propagation equations for solids were investigated thoroughly in [35]. Based on this study, the equations of interest are addressed as follows.

The wave equation in solids can be written for all coordinates as:(3)$$\rho \frac{\partial^{2}u}{\partial t^{2}}=\left(\lambda +\mu \right)grad\;divu+\mu \nabla^{2}u$$

In Equation (3), *t* is time, ρ is density, ∇ is the Laplace operator, u is the displacement vector, *ν* is the Poisson ratio, *E* is Young’s modulus, and *G* is the shear modulus; the Lamé constants are λ=νE/(1+ν)(1−2ν) and μ=G=E/2(1+ν).

It is possible to prove that two types of waves propagate at different speeds in an infinite solid medium. Accordingly, it is possible that any vector field can be presented from the vector analysis as a summation of two vectors—one with a scalar potential and the other with a vector potential:(4)$$u=u_{l}+u_{t}=grad\varphi +rot\psi$$
where φ is the vibration velocity potential, $u_{t}$ is the transversal displacement vector, and $u_{l}$ is the longitudinal displacement vector. It can be noted that $rotu_{l}=divu_{t}=0$. Substituting Equation (4) into Equation (3) and applying the rotation (curl) and divergence operations, the equation converts to:(5)$$\partial^{2}u_{l}/\partial t^{2}-c_{l}^{2}\nabla^{2}u_{l}=0;\;\;\;\;c_{l}=\sqrt{\left(\lambda +2\mu \right)/\rho }$$
and
(6)$$\partial^{2}u_{t}/\partial t^{2}-c_{t}^{2}\nabla^{2}u_{t}=0;\;\;\;\;\;c_{t}=\sqrt{\mu /\rho }$$
where $c_{t}$ is the transversal wave speed and $c_{l}$ is the longitudinal wave speed.

The abovementioned equations describe the propagation of elastic waves for an unbounded solid. However, if the propagation of elastic waves on the solid surface is considered, the equations should be expanded. These are called Rayleigh equations [36] and are obtained by substituting Equation (7) into Equation (5) and Equation (8) into Equation (6): (7)$$u_{l}=Ae^{-\sqrt{k_{s}^{2}-k_{t}^{2}y}}e^{-jk_{s}x}$$
(8)$$u_{t}=Be^{-\sqrt{k_{s}^{2}-k_{t}^{2}y}}e^{-jk_{s}x}$$
where $k_{s}=\omega /c_{s}$ and $\;k_{t}=\omega /c_{t}$. The elastic properties of the materials used in this study are shown in Table 1:

### 2.3. Significant Equations for the Principle of Piezoelectric Sensors

In this study, the piezoelectric principle is used for sensing acoustic waves. In [37], the constitutive and governing equations of this principle for detection and actuation are discussed extensively. Here, we will focus on the equations significant to a piezoelectric sensor. The direct piezoelectric effect, which makes the sensing applications of dynamic signals (including acoustic wave signals) possible, is the ability of some crystalline materials to generate an electrical charge in proportion to the applied mechanical disturbance.

As shown below, the constitutive equations for a sensor made of a one-dimensional piezoelectric material are [37]:(9)$$S=s^{E}T+d_{33}E$$
(10)$$D=d_{33}T+\varepsilon^{T}E$$
where *S* and *D* (Coulomb/m^2^) are the strain and electric displacement, respectively; *s^E^* is the compliance; *T* (N/m^2^) is the stress; *E* (V/m) is the electric field; *d*_33_ (m/V or Coulomb/N) is the piezoelectric constant; and $\varepsilon^{T}$ is the dielectric constant.

Parameters *T* and *E* are independent variables, while *S* and *D* are dependent variables in Equations (9) and (10). If the equations are rearranged such that *E* and *S* are independent and *T* and *D* are dependent, they take the form of:(11)$$T=-\frac{d_{33}}{s^{E}}E+\frac{1}{s^{E}}S$$
(12)$$D=\left(1-\frac{d_{33}^{2}}{s^{E}\varepsilon^{T}}\right)\varepsilon^{T}E+\frac{d_{33}}{s^{E}}S$$
(13)$$T=-e_{33}E+c^{E}S$$
(14)$$D=\left(1-k^{2}\right)\varepsilon^{T}E+e_{33}S$$

In these equations, *e*_33_ = *d*_33_/*s^E^* (C/m^2^) is the product of *d_33_* by the Young modulus, *c^E^ = 1/s^E^* (N/m^2^). The parameter *k* is called the electromechanical coupling factor and can be defined as:(15)$$k^{2}=\frac{d_{33}^{2}}{s^{E}\varepsilon^{T}}=\frac{e_{33}^{2}}{c^{E}\varepsilon^{T}}$$

The conversion of mechanical disturbance to electrical current or applied electric current to mechanical response is evaluated by this parameter. From Equations (13) and (14), the dielectric constant under zero strain can be seen as $\varepsilon^{T}\left(1-k^{2}\right)$. If a sensor is stacked, it is possible to obtain the constitutive equations by integrating Equations (13) and (14) over the volume.
(16)$$\Delta =\frac{1}{K_{a}}f+nd_{33}V$$
(17)$$Q=CV+nd_{33}f$$
(18)$$f=-nd_{33}K_{a}V+K_{a}\Delta$$
(19)$$Q=C\left(1-k^{2}\right)V+nd_{33}K_{a}\Delta$$

The following applies to Equations (17) and (19). *f* is the total force (mechanical disturbance), which is equal to *A*·*T*. *Q* is the total electric charge on the electrodes of the sensor and is equal to *nA·D*—where *n* is the number of elements, and *A* is the cross-sectional area. Δ=Sl is the total extension, where *l = nt* is the sensor length. *C* is the capacitance of the sensor (*C* = $\varepsilon^{T}$·*A*·*n^2^/l*)*,* and *K_a_* is the stiffness (*K_a_ = A/s^E^l*). *V* is the voltage applied between the electrodes of the sensor. The electromechanical coupling factor can be written with the *n*, *K_a,_* and *C* parameters as:(20)$$k^{2}=\frac{d_{33}^{2}}{s^{E}\varepsilon^{T}}=\frac{n^{2}d_{33}^{2}K_{a}}{C}$$

Subsequently, Equations (18) and (19) can be inverted and rewritten as:(21)$$V=\frac{1}{C\left(1-k^{2}\right)}\left(Q-K_{a}nd_{33}\Delta \right)$$
(22)$$f=\frac{K_{a}}{1-k^{2}}\left(-\frac{nd_{33}}{C}Q+\Delta \right)$$

It can be seen from Equations (21) and (22) that the stiffness with open electrodes (*Q*= 0) is *K_a_/1 − k^2^*, and the capacitance for a fixed geometry (Δ=0) is *C*(1 *− k*^2^). For large *k* values, the stiffness changes significantly with the electrical boundary conditions, while the capacitance depends on the mechanical boundary conditions.

If a discrete sensor is subjected to a mechanical disturbance, such as a force with an amplitude of *f*, the total power transmitted to the sensor is the sum of the mechanical ($f\dot{\Delta }$) and electric (*Vi*) power. Thus, the network can be written in the differential form as:(23)$$dW=Vidt+f\dot{\Delta }dt=VdQ+fd\Delta$$

For a conservative element, Equation (23) is converted into a stored energy equation. Equation (24) is obtained by integrating this equation from the reference state to the final state and by substituting the equivalents of *f* and *V* in the corresponding Equations (21) and (22) to obtain the analytical expression of stored electromechanical energy for a piezo sensor.
(24)$$W_{e}\left(\Delta ,Q\right)=\frac{Q^{2}}{2C\left(1-k^{2}\right)}-\frac{nd_{33}K_{a}}{C\left(1-k^{2}\right)}Q\Delta +\frac{K_{a}}{1-k^{2}}\frac{\;\;\Delta^{2}}{2}$$

The first term of Equation (24) is the electrical energy stored in the capacitance $C\left(1-k^{2}\right)$. The second term is the piezoelectric energy, and the last term is the elastic strain energy stored in spring stiffness as $K_{a}/\left(1-k^{2}\right)$.

Up to this point, the equations are defined for a one-dimensional sensor. For more complicated applications, generalized equations must be addressed. Consequently, the constitutive equations of the general piezoelectric material are [37]:(25)$$T_{ij}=c_{ijkl}^{E}S_{kl}-e_{kij}E_{k}$$
(26)$$D_{i}=e_{ikl}S_{kl}+\varepsilon_{ik}^{S}E_{k}$$

In Equations (25) and (26), $\varepsilon_{ik}^{S}$ is the dielectric constant under constant strain, $e_{ikl}$ (Coulomb/m^2^) is the piezoelectric constant, $c_{ijkl}^{E}$ is Hooke’s tensor, $T_{ij}$ is the stress tensor, and $S_{kl}$ is the strain tensor.

In these equations, classic tensor notations are used, in which all indices (*i*, *j*, *k*, *l* = 1, 2, 3, 4) and all repeated indices are summarized. Equations (25) and (26) are the generalized forms of Equations (13) and (14). Similar to the transition from Equations (9) and (10) to Equations (13) and (14), the dependency of the parameters can be shifted, and Equations (25) and (26) can be rewritten as:(27)$$S_{ij}=s_{ijkl}^{E}T_{kl}+d_{kij}E_{k}$$
(28)$$D_{i}=d_{ikl}T_{kl}+\varepsilon_{ik}^{T}E_{k}$$
where $d_{ikl}$ is the piezoelectric constant (Coulomb/N), $\varepsilon_{ik}^{T}$ is the dielectric constant under constant stress, and $s_{ijkl}^{E}$ is the tensor of compliance under a constant electric field.

The required engineering vector notations can be used instead of tensor notations:(29)$$T=\left\{\begin{array}{c} T_{11}\\ T_{22}\\ T_{33}\\ T_{23}\\ T_{31}\\ T_{12} \end{array}\right\}\;\;\;\;\;\;\;\;S=\left\{\begin{array}{c} S_{11}\\ S_{22}\\ S_{33}\\ 2S_{23}\\ 2S_{31}\\ 2S_{12} \end{array}\right\}$$

With these changes, Equations (25)–(28) can be rewritten in matrix form:(30){T}=[c]{S}−[e]{E}
(31)$$\left\{D\right\}={\left[e\right]}^{T}\left\{S\right\}+\left[\varepsilon \right]\left\{E\right\}$$
(32){S}=[s]{T}+[d]{E}
(33)$$\left\{D\right\}={\left[d\right]}^{T}\left\{T\right\}+\left[\varepsilon \right]\left\{E\right\}$$

In Equations (32) and (34), the superscript *T* indicates the transpose of the matrix, and for the sensing applications, the explicit form of Equation (33) is:(34)$$\left\{\begin{array}{c} D_{1}\\ D_{2}\\ D_{3} \end{array}\right\}=\left[\begin{array}{cccccc} 0 & 0 & 0 & 0 & d_{15} & 0\\ 0 & 0 & 0 & d_{24} & 0 & 0\\ d_{31} & d_{32} & d_{33} & 0 & 0 & 0 \end{array}\right]\left\{\begin{array}{c} T_{11}\\ T_{22}\\ T_{33}\\ T_{23}\\ T_{31}\\ T_{12} \end{array}\right\}+\left[\begin{array}{ccc} \varepsilon_{11} & 0 & 0\\ 0 & \varepsilon_{22} & 0\\ 0 & 0 & \varepsilon_{33} \end{array}\right]\;\left\{\begin{array}{c} E_{1}\\ E_{2}\\ E_{3} \end{array}\right\}$$

### 2.4. Experimental Setup

A schematic of the experimental setup is shown in Figure 3a. The experiment is performed on six specimens comprising three materials and two surface treatments for each material: one milled and one ground aluminum specimen; one milled and one ground AISI 1040 specimen; and one lathed and one milled stainless-steel specimen. The geometric forms of the test specimens are rectangular prismatic, and their dimensions are 60 × 60 × 10 mm. The surface roughness measurement device used in the experiment is the Mahr Perthometer M2, and a piezo disk with a diameter of 35 mm is placed on each sample, as shown in Figure 3b. The radius of the profilometer stylus tip used for surface roughness measurements is r_tip_ = 2 μm. The measurement distance ($\lambda_{t}$) of the diamond tip of the Mahr M2 device is set to 17.5 mm; the cutoff distance ($\lambda_{c}$) is set to 2.5 mm. By subtracting the cutoff length $\left(\lambda_{c}\right)$ from the start and end of the total measurement distance ($\lambda_{t}$), the resulting final distance provides the measurement result. The obtained final distance is the one used in the roughness measurement result parameters. For this reason, cutoffs are made proportionally to the first and last parts of the raw sound signal vectors.

The piezo disk is connected to the computer through an AUX cable, and surface roughness profile graphics are digitized. Using Audacity software, the piezo disk is detected as a microphone. While the diamond tip of the Mahr M2 is moved on the specimen to measure the *Ra* and *Rz* parameters, a sound recording was simultaneously initiated, and the friction sound of the diamond tip moving on the specimen was recorded. The main purpose here is to make the sounds of friction and crackling—which are barely perceptible to the human ear—into interpretable signals using the ability of the piezo transducers to detect very low-pitched signals or low amplitude vibrations as certain voltage values.

The Mahr Perthometer M2 surface roughness measuring device performs the roughness measurement by moving the diamond tip forward and backward on the sample. While the forward motion speed is constant, the backward return speed can be adjusted in three stages. For the sound data to be evaluated accurately, the sound occurring at all operating speeds of the device is recorded and analyzed. The sampling frequency for all audio data is 44,100 Hz. Frequency spectrum data of each vector are created separately. Subsequently, the obtained surface roughness profile data, sound recording vector data, and voice recording spectrum data are all imported into the software.

## 3. Results

In the experiments, the *Ra* and *R*z parameters and the surface roughness profiles are obtained from the Mahr M2 device, while the original audio raw signal (*A_rs_*), audio power spectrum (*A_ps_*) parameters, and power spectrum plots are created using the recorded sound data from the piezo acoustic sensor. *A_rs_* and *A_ps_* parameters are calculated from the raw sound data and power spectrum data based on the *Rz* Equation (2). The parameter values are presented in Table 2:

An investigation of the roughness profiles for the aluminum specimens (Figure 4 and Figure 5) showed that there are no sudden fluctuations on the milled surface. However, on the ground surface, although the smoothness of the surface profile seemed to be preserved throughout the measurement length, sudden discontinuities were found in the two small regions. Accordingly, when a parameter comparison is made between the milled surface and the ground surface, the increase in milled surface values is found to be 5.53 times in the *Ra* parameter and 2.56 times in the *Rz* parameter. Therefore, it is confirmed that a more sensitive value is obtained with the *Rz* parameter, and the results obtained in acoustic measurements are compared with this parameter.

The audio power spectrum of the aluminum specimen is provided in Figure 6. From this point on, for all audio recordings, the sampling frequency is 44,100 Hz. The local maxima values for this specimen are observed between 800 Hz and 5000 Hz. Legend abbreviations and corresponding sampling time values are provided in Table 3. If the shortest sampling times are considered during the return stroke of the device tip, the frequency response is largely similar in form for both the milled and ground surfaces, but the amplitudes are different. Additionally, it is determined that the amplitude values of the audio are inversely proportional to the length of the sampling time for all measurements.

An investigation of roughness profiles for the stainless-steel specimens (Figure 7 and Figure 8) shows that no significant discontinuity is detected on either the milled or lathed surfaces. However, a small deviation and an increase in profile amplitude are observed from the 4 mm point on the milled surface. Thus, compared to the 3.459 times increase in *Ra* values, 4.201 times increase in the value of *Rz* is determined. Therefore, it can be deduced that the sensitivity of the parameters does not create a serious issue for these specimens.

The audio power spectrum of the stainless-steel specimen is provided in Figure 9. The variation between the milled surface and lathed surface values appears as a shift in the frequency spectrum. The frequency response for the milled surface shows two steps between 700–2000 Hz and 2000–3200 Hz; for the lathed surfaces, there are three steps between 700–1250 Hz, 1250–2000 Hz, and 2000–3200 Hz. The local maxima, as the signal strength for the lathed surfaces, are between 700–1250 Hz.

An investigation of roughness profiles for the AISI 1040 specimens (Figure 10 and Figure 11) shows that considerable discontinuities are detected on both the milled surface and the ground surfaces in terms of surface quality. This situation creates large variations between *Ra* and *Rz* values, especially for the same machining operation. When considering these differences, it was determined that the *Ra* parameter, in particular, does not provide sensitive values for this specimen.

The audio power spectrum of the AISI 1040 specimen is provided in Figure 12. The frequency response for the ground surface spans a wide bandwidth and is observed between 700–1200 Hz, 1200–2000 Hz, and 2000–3000 Hz. The frequency response for the milled surface shows a unique amplitude spike, unlike other specimens, at approximately 7000–8000 Hz.

## 4. Discussion

Piezo acoustic data obtained for all samples and production operations were evaluated in three stages: a parametric analysis, a power spectrum analysis, and finally, a comparative analysis with standard parameters. The RMS and *A_rs_* acoustic surface roughness parameters obtained within the scope of the first stage provided accurate information, especially about the surface quality obtained with different production methods of the same material. These parameters guided the other stages by taking lower values on the smoother surface for all materials. Then, sound power spectrum graphs, including all sampling rates, materials, and production methods, were produced. The power spectrum of the aluminum sample showed a frequency response compatible with the surface roughness profile obtained by standard measurements of the same material. In this graph, it is determined that the smoother surface creates a low-intensity signal strength and the signal strength increases when the surface roughness increases. When stainless-steel samples are examined, it is seen that the greatest difference between the lathed and milled surfaces is the difference in frequency responses. It is clear that this situation is due to the very tight formation of the hills and pits, especially on the lathed surface. Due to the aforementioned tight formation, the timbre and the frequency of the sound were increased in the sound recordings taken from that surface, and this was determined as the frequency shift in the power spectrum graph. This result was shown in the power spectrum graph together with the roughness profile graphs, where the surface quality of the AISI 1040 material is problematic. Unlike other materials, the surface condition showed sudden “spikes”, even around 7000 Hz. When considering all these results, it can be seen that an examination of the power-dependent power spectrum graphs will provide as much information as the profiles obtained by the traditional method, including the information related to the maximum and minimum values of the peak and trough values, as well as the frequency axes in the power spectrum graphs. It has been observed that it can be obtained by working on it. The *A_ps_* value obtained from the sound power data of the system also obtained a numerical value comparable with the *Rz* value among all materials depending on the correction parameters, the material density, and the surface speeds, especially for the horizontal material on the horizontal axis, whether compact compression for stainless steel or irregular compression for AISI 1040. A surface roughness value with high sensitivity, including spreading information, has been reached.

## Figures and Tables

**Figure 1 sensors-22-04381-f001:**
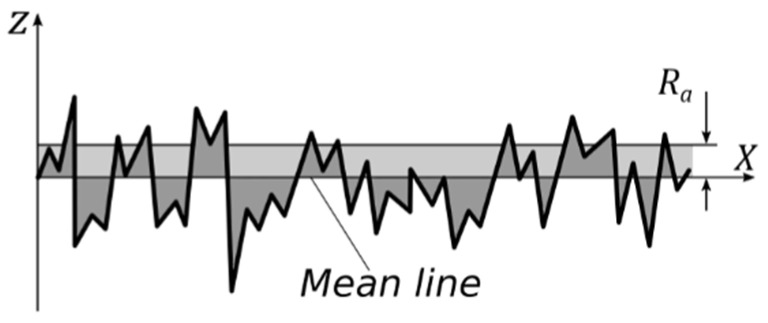
Average absolute value (*Ra*).

**Figure 2 sensors-22-04381-f002:**
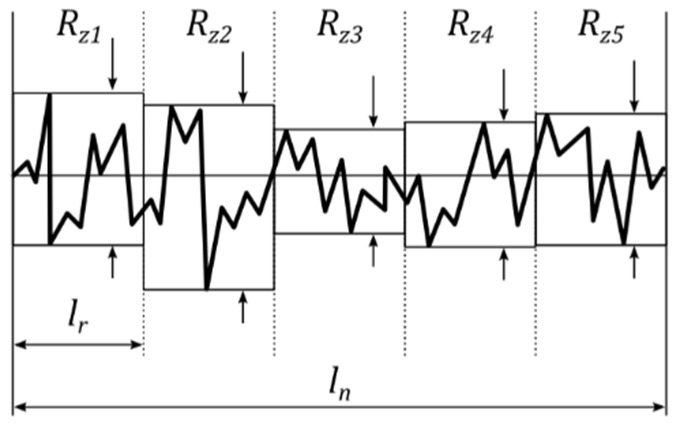
Average profile height (*Rz*).

**Figure 3 sensors-22-04381-f003:**
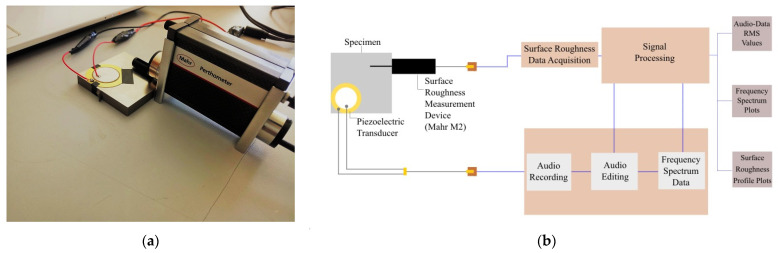
Photograph (**a**) and schematic (**b**) of the experimental setup.

**Figure 4 sensors-22-04381-f004:**
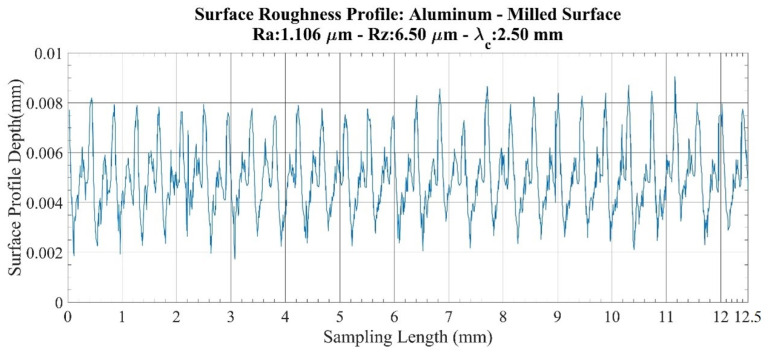
Surface roughness profile: aluminum–milled surface.

**Figure 5 sensors-22-04381-f005:**
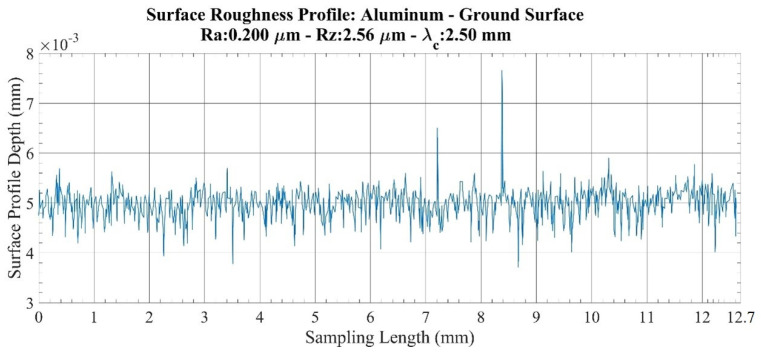
Surface roughness profile: Aluminum-ground surface.

**Figure 6 sensors-22-04381-f006:**
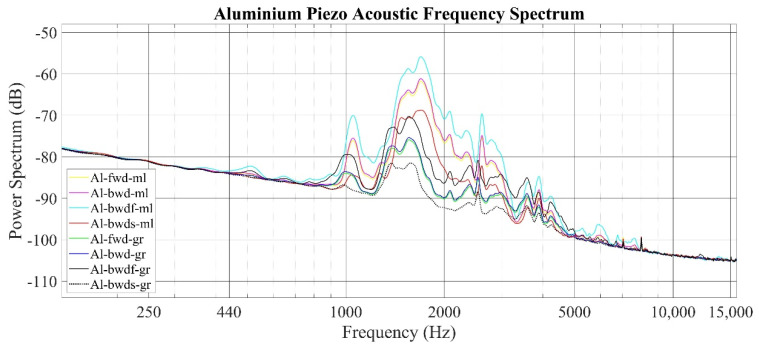
Piezo acoustic measurement frequency spectrum of the aluminum specimen (frequency resolution: 50 Hz).

**Figure 7 sensors-22-04381-f007:**
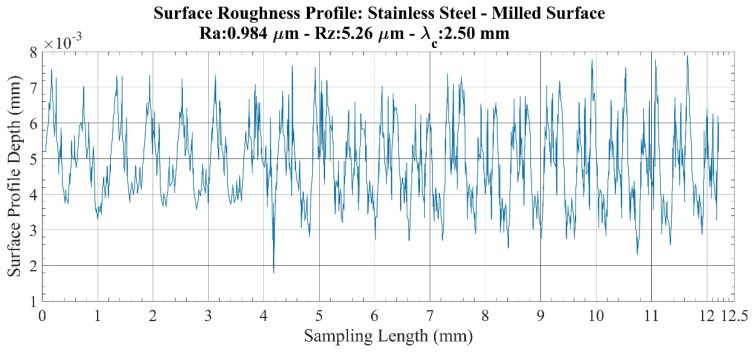
Surface roughness profile: Stainless steel-milled surface.

**Figure 8 sensors-22-04381-f008:**
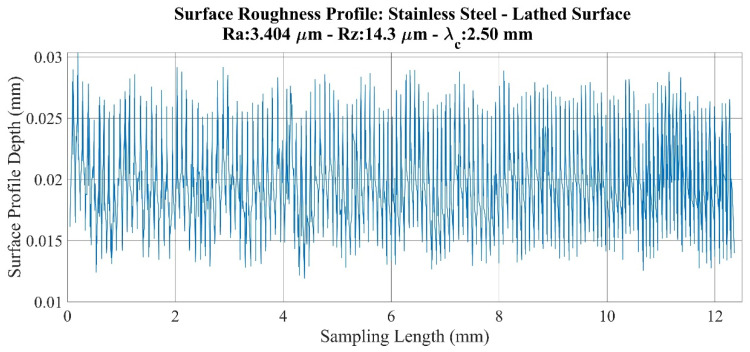
Surface roughness profile: Stainless steel–lathed surface.

**Figure 9 sensors-22-04381-f009:**
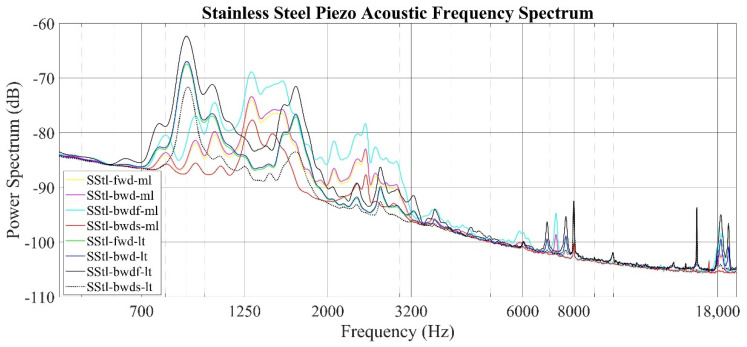
Piezo acoustic measurement frequency spectrum of the stainless steel specimen (frequency resolution: 50 Hz).

**Figure 10 sensors-22-04381-f010:**
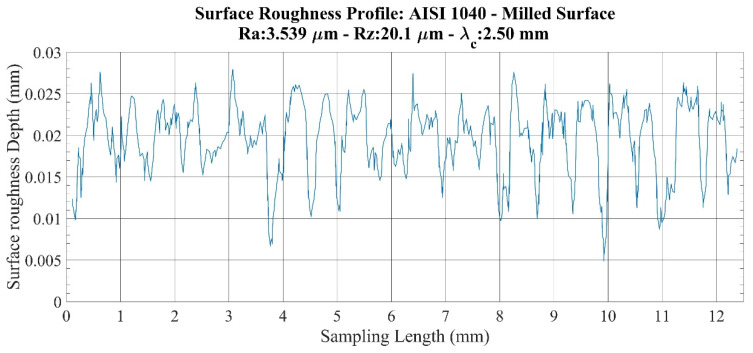
Surface roughness profile: AISI 1040–milled surface.

**Figure 11 sensors-22-04381-f011:**
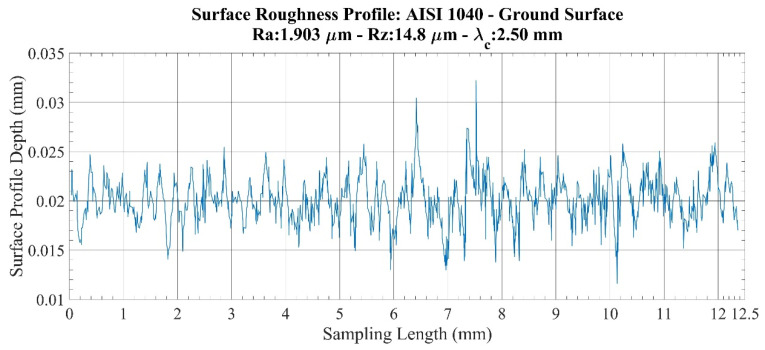
Surface roughness profile: AISI 1040–ground surface.

**Figure 12 sensors-22-04381-f012:**
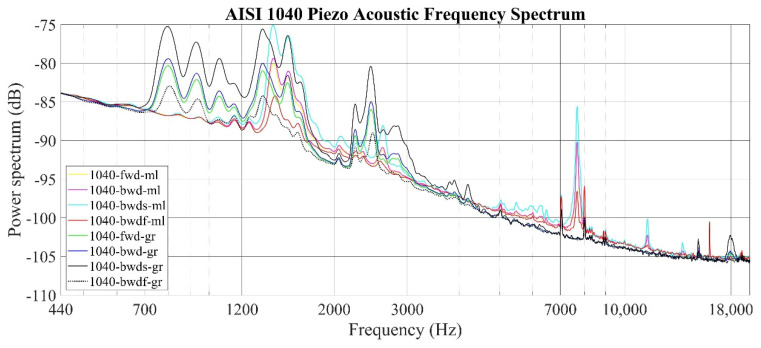
Piezo acoustic measurement frequency spectrum of the AISI 1040 specimen (600–10,000 Hz) (Frequency Resolution: 50 Hz).

**Table 1 sensors-22-04381-t001:** Elastic properties of specimen materials [35].

Material	$\mathrm{Density}\;\rho ,\;10^{3}kg/m^{3}$	$\mathrm{Velocity}\;\mathrm{of}\;\mathrm{Wave}\;\mathrm{Propagation}\;c,\;10^{3}m/s$		
		Longitudinal	Transversal	Surface
Aluminum	2.7	6.35	3.08	2.80
Stainless Steel	8.03	5.73	3.12	2.90
AISI 1040	7.80	5.92	3.28	3.01

**Table 2 sensors-22-04381-t002:** *RMS, A_rs_, A_ps,_* values of acoustic measurements; *Ra* and *Rz* values of surface roughness measurements.

	Aluminum (Milled)	Aluminum (Ground)	Stainless Steel (Milled)	Stainless Steel (Lathed)	AISI 1040 (Milled)	AISI 1040 (Ground)
Sound *RMS*	3.6027	1.0279	1.2262	1.3868	0.8783	0.8085
*Ra* (μm)	1.106	0.20	0.984	3.404	3.359	1.903
*Rz* (μm)	6.50	2.56	5.26	14.30	20.10	14.80
*A_rs_*	28.3730	8.0338	27.5645	33.7739	19.8591	18.3818
*A_ps_*	11.2109	9.4554	30.3371	39.7898	32.8009	24.7656

**Table 3 sensors-22-04381-t003:** Power spectrum plot legend abbreviations and corresponding sampling time values.

Legend	Specimen	Stroke-Speed	Machining Operation	Sampling Time, Sec
Al-fwd-ml	Aluminum	Forward–Standard	Milling	55
Al-bwd-ml	Aluminum	Backward–Standard	Milling	48
Al-bwdf-ml	Aluminum	Backward–Fast	Milling	12.3
Al-bwds-ml	Aluminum	Backward–Slow	Milling	35.3
Al-fwd-gr	Aluminum	Forward–Standard	Grinding	55
Al-bwd-gr	Aluminum	Backward–Standard	Grinding	47.5
Al-bwdf-gr	Aluminum	Backward–Fast	Grinding	12
Al-bwds-gr	Aluminum	Backward–Slow	Grinding	35.4
SStl-fwd-ml	Stainless Steel	Forward–Standard	Milling	56.5
SStl-bwd-ml	Stainless Steel	Backward–Standard	Milling	48
SStl-bwdf-ml	Stainless Steel	Backward–Fast	Milling	12.3
SStl-bwds-ml	Stainless Steel	Backward–Slow	Milling	35.7
SStl-fwd-lt	Stainless Steel	Forward–Standard	Lathing	52.5
SStl-bwd-lt	Stainless Steel	Backward–Standard	Lathing	46
SStl-bwdf-lt	Stainless Steel	Backward–Fast	Lathing	12
SStl-bwds-lt	Stainless Steel	Backward–Slow	Lathing	34
1040-fwd-ml	AISI 1040	Forward–Standard	Milling	60
1040-bwd-ml	AISI 1040	Backward–Standard	Milling	50
1040-bwdf-ml	AISI 1040	Backward–Fast	Milling	14.5
1040-bwds-ml	AISI 1040	Backward–Slow	Milling	35.5
1040-fwd-gr	AISI 1040	Forward–Standard	Grinding	62
1040-bwd-gr	AISI 1040	Backward–Standard	Grinding	48
1040-bwdf-gr	AISI 1040	Backward–Fast	Grinding	12.5
1040-bwds-gr	AISI 1040	Backward–Slow	Grinding	35.5

## Data Availability

The data presented in this study are available on request from the corresponding author.

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
