# Peer review of "Utilizing Piezo Acoustic Sensors for the Identification of Surface Roughness and Textures"

_sensors, 2022, doi:10.3390/s22124381_

Round 1
Reviewer 1 Report
This paper is focused on exploring and utilizing piezo acoustic sensors for the correlation between acoustic measurements and surface roughness measurements. The mathematical apparatus of the sensor very well written and explained in detail. It has shown that this method is easy to use and quite accurate in determining the roughness of surfaces with good resolution. Three-stage evaluation of all samples and production operations presented. According to the presented results, it can be seen that an examination of the power-dependent power spectrum graphs will provide as much information as the profiles obtained by the traditional method. The authors showed that with presented method it is possible to measure the surface roughness value with high sensitivity.
The work is interesting and worthy of publication in the J. Sensors.
However, there are small notes that need to be corrected.
1) Equation 6 do not has an equality, which makes it unsolvable.
2) Equations 7 and 8 are the same – (if it is that mistype should be corrected, one it has any porpoise, please highlight it)
3) Line 273 – “Figure 9. Piezoacoustic measurement frequency spectrum of the AISI 1040 specimen (frequency resolution: 50 Hz)” while the link above is referred to “The audio power spectrum of the stainless-steel specimen is given in Fig. 9.” Line 264.
4) Was there any selective filter or noise isolation used?
Author Response
Dear Respected Reviewer,
On behalf of my coauthors, we sincerely appreciate your comments and feedback about our manuscript sensors-1734772”, entitled “Utilizing piezo acoustic sensors for the identification of surface roughness and textures”.
Our detailed responses to your comments and relevant revisions are attached

Reviewer 2 Report
In attachment file.

Author Response

(The authors gave the same response as above.)

Round 2
Reviewer 2 Report
1. I appreciate the reference in the answer to my previous comments to the concept of friction sound. This phenomenon is very complex as can be seen in the article "Akay A. Acoustics of friction. J Acoust Soc Am. 2002 Apr;111(4):1525-48. doi: 10.1121/1.1456514. PMID: 12002837" being dependent not only on the mechanical properties of the surfaces in contact !? (surface roughness and textures).
2. The response of any type of sound sensor after calibration/linearization can be considered viable in some limit - your sensor is 10 cents buzzer used in toys. Unfortunately, this does not mean that this stage (calibration) has been completed. It is a commendable attempt to convince readers of a Q1 journal about this situation by describing the calibration procedures and making a demonstration using different types of sound sensors in your proposed experiment without changing the result.
3. In my opinion, a new method must be able to measure at least with equal precision the parameters of interest. In this sense, the logical expectation is to have a table or graph in which to see a comparative study between a standard method (contact or non-contact one) and this new method. In this way a demonstration of what the new method can do in relation to one or more traditional methods would have avoided a complex of explanations based on references in an indirect incident with the proposed method.
4. I have seen with this buzzer for toys or from toys many articles starting with scanning tunneling microscope, adaptive optics, energy recovery, many interferometers, ... and all application presented in literature are true, and these applications can be appreciated for teaching purposes, and it is too much to consider such demonstrations around buzzers and standard PC sound card and open source audio editor as a solution equivalent to professional approaches.